# Hand contamination and hand hygiene knowledge and practices among commercial transport users after the SARS-CoV-2 virus (COVID-19) scare, Enugu State, Nigeria

Ifeanyi O. Aguzie[1]*, Ahaoma M. Obioha[1], Chisom E. Unachukwu[1], Onyekachi J. Okpasuo[1], Toochukwu J. Anunobi[2], Kenneth O. Ugwu[3], Patience O. Ubachukwu[1], Uju M. E. Dibua[3]

1 Department of Zoology and Environmental Biology, University of Nigeria, Nsukka, Nigeria, 2 Department of Science Laboratory Technology, Federal Polytechnic, Idah, Kogi State, Nigeria, 3 Department of Microbiology, University of Nigeria, Nsukka, Nigeria

☙ These authors contributed equally to this work.
* ifeanyi.aguzie@unn.edu.ng

**Data Availability Statement:** Data is publicly available in data repository, and freely accessible:

## Abstract

Contaminated hands are one of the most common modes of microorganism transmission that are responsible for many associated infections in healthcare, food industries, and public places such as transportation parks. Public health approaches during COVID-19 pandemic have shown that hand hygiene practices and associated knowledge are critical measure to control the spread of infectious agent. Hence, assessment of commercial transport users' knowledge, belief and practices on hand hygiene, and potential contamination with infectious agents which is the aim of the study, aligns with general health concern of quantifying contamination risk levels to predict disease outbreaks. This study utilized a randomized sampling approach to select 10 frequently used commercial parks within two districts in the State: Enugu and Nsukka. The parameters analysed include a cross-sectional questionnaire survey, hand swab and hand washed samples collected from dominant hand of participants. A total of 600 participants responded to the questionnaire survey, while 100 participants' hand swabs were examined for microbial contamination. This study recorded a high prevalence of fungal (90.0%) and bacterial (87.0%) species; 20 species of fungus were identified with prevalence range of 1% to 14%; 21 bacterial species were isolated with prevalence range of 1% to 16%. These species were identified as either opportunistic, non-invasive, or pathogenic, which may constitute a health concern amongst immunocompromised individuals within the population. *Aspergillus* spp. (14%), was the most common fungal species that was exclusively found amongst Nsukka commercial users, while *E. coli* was the most prevalent isolated bacterial species amongst Nsukka (12%) and Enugu (20%) commercial park users. Prevalence of fungal contamination in Nsukka (94.0%; 47/50) and Enugu (86.0%; 43/50) were both high. Prevalence of bacterial contamination was higher in Enugu than Nsukka but not significantly (47[94.0%] vs. 40[80.0%], p = 0.583). A greater number of participants (99.3%) were aware of the importance of hand hygiene, however with low compliance rate aside "after using the toilet" (80%) and "before eating" (90%), other

Aguzie, Ifeanyi (2024). Hand contamination and hand hygiene knowledge and practices among commercial transport users, Enugu State, Nigeria [Dataset]. Dryad. https://doi.org/10.5061/dryad.h9w0vt4q9.

**Funding:** The authors received no specific funding for this work.

**Competing interests:** The authors have declared that no competing interests exist.

relevant hand washing and sanitizing practices were considered less important. With these observations, we can emphatically say that despite the COVID-19 scare, commercial park users within the sampled population do not efficiently practice quality hand wash and hygiene measures, hence, risking the widespread of infectious agents in situation of disease outbreak or among immunocompromised individuals.

## Introduction

The structure and orientation of the human body have equipped the hand with a variety of functions that it can perform extremely gentle and precise actions involving touching, grasping, feeling, holding, manipulating, caressing, and more. These functions involve day-to-day interaction with objects and exposed surfaces which harbour harmful pathogens, fomites, and substances that contaminate the hands with great potentials for causing infections. Hence contaminated hands are one of the most common modes of microorganism transmission [1]. These disease-causing infection transmissions are caused by transient flora which are located superficially on the skin and could be readily transmitted to the next thing that is touched. They are responsible for many healthcare-associated infections (HAIs) in healthcare institutions, also in food industries, public places such as public transport parks, and domestic gatherings. These pathogens are easily removed by proper hand washing [2].

Hand hygiene which involves cleaning one's hands to remove dirt, grease, and potential pathogens such as bacteria, viruses, parasites, and unwanted substances through handwashing with soap and water or the use of 60% alcohol-based hand sanitizers is an important public health safety procedure. Hand hygiene practice is essential in the prevention and transmission of infectious diseases at home and in everyday life [3,4]. It is the cornerstone and starting point of infection control and prevention [5]. Failure to perform appropriate hand hygiene practices is a leading cause of HAIs and the spread of multi-resistant organisms and a significant contributor to outbreak of infectious diseases [6,7]. The outbreak of the COVID-19 pandemic has recently drawn unprecedented attention to hand hygiene. The pandemic has shone a light on hand hygiene as an inexpensive, widely applicable protective measure, which has been recognized as part of a multicomponent public health approach that includes wearing a face mask and physical separation [8].

The WHO, 'Save Lives: Clean your Hands' campaign acknowledged the importance of hand hygiene by stating that, 'Hand hygiene is one of the most effective actions you can take to reduce the spread of pathogens and prevent infection as well as the COVID-19 virus' [8]. Hand hygiene has thus been at the forefront of many countries' responses to tackling the COVID-19 pandemic. Similarly, The United States Centers for Disease Control and Prevention (CDC) has highlighted the role of hand hygiene in preventing the spread of infection, saying: 'Hand hygiene helps stop the spread of germs, including ones that can cause antibiotic-resistant infections which can defeat the drugs designed to kill them making them difficult, and sometimes impossible to treat [9]. Moreover, a substantial amount of peer-reviewed literature has shown the benefits of hand hygiene for preventing many infectious diseases including gastrointestinal illnesses [10,11], trachoma and soil-transmitted helminth infections [12], as well as respiratory infections [10]. Research has shown that a lot of bacteria and viruses can grow on a contaminated hand and can help in the spread of diseases such as diarrhoea, staphylococcus, influenza, corona virus and several other acute respiratory infections when self-inoculated [13]. A behavioural observational study conducted by Kwok et al. [14] showed that an

average person touches his face about 2,346 times in 240 minutes, showing that contaminated hands have a high chance of transmitting infections to the face. Findings from meta-analysis suggests that hand washing with soap can reduce respiratory infections by between 21% and 23%, and diarrheal disease by between 23% and 48% [15]. Thus, hand hygiene practices are critical not only during a pandemic but also in preventing the spread of other diseases.

Despite institutional policies and protocols on hand washing, there is still a poor hand hygiene compliance amongst individuals including commercial drivers and passengers. Several observations have been made on this group of people in parks across Enugu State, Nigeria, and it can be inferred that hand hygiene is at a low level even during the waning phase of the COVID-19 scare, because of the general malaise towards the spread of the SARS-CoV-2 virus. Hence, it is crucial that commercial transport users hand hygiene knowledge, belief and practices be assessed to determine whether there is need to enlighten them towards changing their hand hygiene practices. Commercial parks are always crowded with travellers encountering one another and with shared surfaces. Therefore, the rate of contamination in commercial transport parks may be very high if proper hygiene measures are not in place. This study sought to provide concrete information on hand contamination level and hand hygiene knowledge, attitude and practices among commercial park users in Nsukka and Enugu, and to compare the contamination level between the two cities. No such study existed for this demographic in Enugu State from literature search. The results of this study can be used to evaluate and quantify the contamination risk levels and predict future disease outbreaks.

## Materials and methods

### Study area

The study was conducted in Enugu town (06˚30' N– 06˚40' N, 07˚20' E– 07˚35' E) and Nsukka (6˚ 43' N– 7˚ 00' N, 7˚ 13' E– 7˚ 35' E), two major towns in Enugu State, Nigeria. Enugu town has a robust economy, including public and private institutions and businesses with high concentration of transport parks, hotels, banks, recreational facilities, and restaurants. The last national census conducted in Nigeria in 2006 estimated that the population of Enugu town was 722,664, at a national population growth rate of 2.8% [16]. Nsukka is one of the seventeen LGAs that make up Enugu State. The population of Nsukka was estimated to be 309,448 in 2006 [16].

Enugu State belongs to the humid tropical region; it lies in the transitional-Savannah region; derived from prolonged cultivation. Average monthly temperature fluctuate between 24˚C and 29˚C, and is usually a little above 27˚C all through the year, although it sometimes exhibits peak of up to 36˚C in March, which is usually the hottest month of any year. Average annual rainfall is about 1800 mm but over 70% of the amounts fall in four months, between June and September [17]. A typical day at the commercial parks in Enugu town and Nsukka would show passengers struggling to get into buses, having very close contact with each other, everyone touching almost the same surface, using the same pen to write on the travel manifest.

### Selection of sampling site

Enugu town is made of three LGAs namely Enugu East LGA, Enugu North LGA, and Enugu South LGA [18]. Given that the target population is commercial transport users, five transport parks were randomly selected in Enugu town, and five in Nsukka (Table 1). A simple random lottery method sampling was used to select the parks.

**Table 1. Sampled parks and coordinates in Enugu town and Nsukka, Enugu State, Nigeria.**

| Park | | Location | LGA | Latitude | Longitude |
|---|---|---|---|---|---|
| 1 | Peace Mass Transit (EN) | Holy-ghost | Enugu North | 6° 26' 16.5" N (6.43791) | 7° 29' 22.5" E (7.489581) |
| 2 | Royal Mass Transit | Achara | Enugu North | 6° 26' 17.0" N (6.438043) | 7° 29' 16.4" E (7.487891) |
| 3 | Entranco Iwollo-oye | Abakpa | Enugu East | 6° 28' 57.4" N (6.482601) | 7° 30' 52.3" E (7.514540) |
| 4 | Eastern Mass Transit | Coal-camp | Enugu North | 6° 26' 02.3" N (6.450651) | 7° 28' 53.1" E (7.481409) |
| 5 | GUO Transport | Achara | Enugu North | 6° 26' 11.4" N (6.436502) | 7° 29' 29.0" E (7.491392) |
| 6 | Opi Com. Mass Park | Opi | Nsukka | 6° 46' 28.1" N (6.77448) | 7° 25' 58.7" E (7.43297) |
| 7 | Nkwo Ibeagwa Park | Ibeagwa-Aka | Nsukka | 6° 54' 56.1" N (6.91557) | 7° 23' 52.6" E (7.39794) |
| 8 | Peace Mass Transit (NSK) | Owere Nsukka | Nsukka | 6° 50' 56.1" N (6.84891) | 7° 23' 53.5" E (7.39820) |
| 9 | Ogige Shuttle Park | Owere Nsukka | Nsukka | 6° 50' 56.9" N (6.84914) | 7° 23' 53.2" E (7.39811) |
| 10 | Royal Mass Transit (NSK) | Obollo | Nsukka | 6° 54' 58.1" N (6.91612) | 7° 30' 42.5" E (7.51180) |

## Sample size determination

The sample size was determined using the equation developed by Cochran [19], a commonly used approach for computing the sample size for a large target population [20]. Using the formula, the sample size (N) was estimated:

$$N = \frac{Z \times P(1 - P)}{d^2}$$

In the absence of previous reports on hand hygiene knowledge, attitude and practice among commercial transport users in Enugu State, the proportion $P$ was set as 0.5, the margin of error ($d$) as 0.05, and Z as 1.96. The estimated sample size was 384. A 10% non-response was anticipated bring the estimated sample size for questionnaire respondents to 422. A hand swap and hand wash sample size of 100 participants (50 each from Nsukka and Enugu) was assumed to have adequate statistical power (80%) at 5% level of significance for microbial diversity studies [21,22].

## Inclusion and non-inclusion criteria

A commercial transport user was defined as a driver or passenger found within the sampled parks at the time of the study. Anybody within this category who volunteered to participate in the study was included. However, a commercial transport users who qualified to participate by all standards except age, below the age of 10 years at the time of sample collection, was excluded; also those between 10 and 14 years whose parents/guardian did not give their consent were not included.

## Sources of data

The sources of data used for this study comprise the hand swab and hand wash samples collected from the dominant hand of drivers and passengers, and the responses to the administered questionnaire. No secondary data was used.

## Ethics statement

Ethical clearance for the study was obtained from the Enugu State Ministry of Health Ethical Review Department (MH/MSD/REC21/259 and MH/MSD/REC21/258). Informed consent was obtained from participants after explanation of the purpose of the study and the procedure of sample collection. Informed consent was given verbally, written or both according to

participants' preferences. However, most (~ 98%) preferably consented verbally. Consent for 1.8% of the respondents who were between 12 and 14 years, required additional approval from their guardian/parents who were with them at the time of sample collection. The study procedure posed no or minimal risk to participants, thus matured adolescents of 15 to 17 who consented were allowed to participate without obtaining additional parental/guardian assent as approved by the ethical committee. The Child Right Act in Nigeria provides that mature adolescent has the right to give consent for scientific investigation without parental consent; even in national HIV prevalence survey 15 years old adolescent are regularly engage [23,24]. The study was conducted strictly following the standard conditions for the ethical approval.

## Questionnaire design

A pretested, descriptive, cross-sectional questionnaire survey design was used to probe the knowledge and perception about hand hygiene, self-reported hand hygiene practices, and beliefs regarding hand hygiene practices among commercial transport users in the selected parks. The questionnaire was divided into four key sections: demographics, knowledge, self-reported practices, and beliefs. The questionnaire consists of 24 items excluding the demographics. The contents were written in English.

## Procurement and preparation of media

All media used in this study were purchased from the University of Nigeria Nsukka (UNN) Chemical store. Media used for the study included: Nutrient agar, MacConkey agar, and Sabouraud dextrose agar (SDA) (TM media in Delhi, India); Urea agar base (Christensen) Autoclavable, Simmon Citrate agar, Triple sugar iron agar (TSI), and SIM medium (Hi Media in Mumbai, India). Nutrient, MacConkey agar and SDA agar are microbiological media for bacteria and fungi growth. Simmon citrate agar, urea agar base, triple sugar iron (TSI) agar and SIM media were all used for biochemical tests on microbes. The preparation of microbial media was done according to the manufacturer's instructions.

## Sample collection

Randomization of participants was done by selecting every third commercial transport user that walked past a member of the research team that was positioned at the main entry into the parks. However, at some parks, it was difficult to identify which was the main entry point as there were multiple entry channels. In such places, the sentinel was positioned at a busy point within the park where park users were likely to pass through before boarding a vehicle. Hand swab samples were collected from 100 commercial transport users, 10 participants from each of the selected parks. Sterile swab sticks (MandeLab Co. Ltd, Shenzhen, China) moistened with 0.1% peptone water were used to take swab samples from the surfaces of the participants' palms, specifically on their dominant hand [25]. Also using plastic collection tubes with lid, hand washed samples were taken from only 22 participants due to low consent towards having their hand washed. The sampling was done in the afternoon between 12:00 PM– 4:00 PM, 1st April to 20th November 2022. The sampled individuals were randomly chosen, and only those who consented to the study and met the inclusion criteria were sampled. The hand wash, and hand swap samples were placed in the cooling container to preserve the collected swabbed samples and were transported for processing to Classic Biomedical Laboratory, Nsukka, Nigeria. Data relating to personal hygiene knowledge, belief and practices were collected by the use of a structured questionnaire.

## Examination of hand swab samples

Swab samples were inoculated onto Nutrient, MacConkey and SDA agar plates using a wire loop flamed with a Bunsen burner for sterility. Nutrient and MacConkey agar were both used to culture bacteria, and SDA agar was used to culture fungi. Chloramphenicol antibiotics were regularly introduced during any preparation of SDA agar to inhibit bacterial growth and enable the growth of fungi microbes only.

After the inoculation of the swab samples onto the microbial agar media, the media plates were sealed in sterile transparent nylon and placed inside the incubator for proper conditioning at a constant temperature of 28˚C to facilitate the growth of bacteria and fungi microbes and prevent contamination [26]. The media plates were observed for the growth of microbes after 24 h. Growth of bacteria was observed in some nutrient and MacConkey media plates. The growing bacteria colonies obtained from the nutrient and MacConkey media were sub-cultured onto new nutrient agar media to get the purest sample and re-incubated for 24 h, after which bacteria growth found in the sub-culture was stocked inside 15 ml Bijou bottles containing nutrient agar and preserved in the refrigerator as reserve for laboratory observation, identification and analysis if needed.

After an incubation period of 4 days, clear fungal growth was observed on the SDA media plates. Isolates of multiple fungal growths on a single SDA media plate were inoculated onto separate plates. The isolates were distinguished based on observation of morphological differences, and were separated to prevent contamination of the culture with one another, prevent growth inhibition, and to enable easy identification of culture samples.

## Examination of hand wash samples

Hand wash samples were examined for the presence of parasites. The 22 hand wash samples were each transferred into test tubes, centrifuged at 3000 rpm for 5 min [27]. The decant sample was placed on a slide and observed under a light microscope (Ningbo Beilun Aofusen instrument, Zhejiang, China) at x40 magnification for the presence of any protozoan or helminth parasite.

## Identification of contaminating microbes

Identification of fungi was done using the slide culture technique coupled with lactophenol cotton blue staining [28], a rapid method for preparing fungal colonies for examination and identification. Identification of bacterial microbes was done using various biochemical test which collectively aided in identifying different kinds of bacteria. The biochemical tests include: Simmon Citrate Test, Urease Test, Catalase Test, SIM test, and TSI test. Catalase test is used to identify organisms that produce the enzyme, catalase. This enzyme detoxifies hydrogen peroxide by breaking it down into water and oxygen gas. The bubbles resulting from the production of oxygen gas indicate a catalase-positive result [29]. The citrate test screens a bacterial isolate for the ability to utilize citrate as its alkaline by-product of citrate metabolism. A positive diagnostic test rests on the generation of the medium is demonstrated by the colour change of a pH indicator. The citrate test is used to identify gram-negative pathogens and environmental isolates [30]. The urease test identifies those organisms that are capable of hydrolyzing urea to produce ammonia and carbon dioxide. It is used as a diagnostic tool for the identification of certain microorganisms, particularly those in the genera *Proteus*, *Klebsiella*, *and Helicobacter* [31]. The SIM (Sulfide, Indole, Motility) test is a microbiological test that is used to identify certain gram-negative bacteria, such as *Escherichia coli* and *Salmonella* [32]. SIM test was used to determine the ability of the bacterial growth to reduce sulphur by producing hydrogen sulphide ($H_2S$). TSI (triple sugar iron) test is a laboratory test that is used to

identify and differentiate between various types of bacteria based on their metabolic reactions. It can be used to determine the presence of glucose, lactose, and sucrose fermentation, as well as hydrogen sulfide production [33].

## Statistical analysis

Data analysis was done in R version 4.2.0 [34] and SPSS version 23.0 (IBM Corporation, Armonk, USA). Chi-square test was used to estimate prevalence of microbial contamination, and its disparities between the two districts and ten parks. Heat maps of prevalence of microbial isolates by motor parks were generated using *stringr* [35] and *ggplot2* [36] packages. Questionnaire responses were summarised as frequency distribution. Five hand hygiene belief items and eights hand hygiene practice items (four habitual and four purposive) making a total of 13 items were subjected to principal component analysis (PCA) in order to generate composite measures of "belief", "habitual practice" and "purposive practice" [37–39]. The PCA which was Varimax rotated generated four principal components (PC) which explained 63.9% of the total variance in hand hygiene belief and practice (S1 Table). PC1 comprising four variables was interpreted to represent "purposive/deliberate hand hygiene practice", PC2 as "belief", and PC3 as habitual/customary hand hygiene practice (S1 Table). The factor score loading of PC1 was used to generate a binary variable by 50 percentile dichotomization of the factor score loading and values below and above the median value coded as "0" for "poor" and "1" for "good" purposive practices respectively. The 50 percentile dichotomy was based on the assumption that purposive hand hygiene practice was uniformly distributed in the population [40], and this assumption was further supported by the almost complete uniform distribution of the factor score loading of PC1. The generated purposive practice variable "practice p50" performed well on cross-tabulation with the four purposive practice variables (S2 Table). Estimates of crude odd ratio (OR) of good purposive/deliberate hand hygiene practice in relation to demographic variables were obtained by binary logistic regression. The level of significance was set at $p \leq 0.05$.

## Results

### Demographics

A total of 600 commercial transport users, 300 each from Nsukka and Enugu town, responded to the questionnaire. Their demographics are presented in Table 2. More than 40% of the respondents, 243 (40.5%), were 20–29 years old. Males were more than twice the number of females, 417 (69.5%) vs. 183 (30.5%). Only 54 (9.0%) of the individuals were identified as not having a formal education. More than one-half had a tertiary education, 336 (56.0%), while 161 (26.8%) had secondary education, and 49 (8.2%) primary education.

### Microbial contamination

Hand swab samples were done for a total of 100 participants (50 each from Nsukka and Enugu). Fungi isolates were present in 47 (94.0%) and 43 (86.0%) of commercial transport users' palms in Nsukka and Enugu respectively, constituting 90% prevalence of fungi species overall. In total, 21 fungi species including two which were difficult to identify were isolated (**Fig 1**). *Aspergillus* spp. (14.0%), *Malassazia* spp. (14.0%), *Mucor* spp. (13.0%), and *Candida albicans* (11.0%), were the four most prevalent fungi species. Some of the fungi species are pathogenic in human. The within districts difference in fungi species prevalence was significant for both Nsukka and Enugu (p < 0.0001). Fungi species isolates in Nsukka and Enugu were 14 (66.7%) and 11 (52.4%) of the 21 species respectively, with only 4 (19.1%) common to

**Table 2. Demographics of questionnaire respondents commercial transport users of motor parks in Nsukka and Enugu towns, Enugu State, Nigeria (*n* = 600).**

| Demographics | Options | Frequency (%) |
|---|---|---|
| Age (years) | 10–19 | 120 (20.0) |
| | 20–29 | 243 (40.5) |
| | 30–39 | 140 (23.3) |
| | $\geq$ 40 | 97 (16.2) |
| | | **600 (100)** |
| Sex | Male | 417 (69.5) |
| | Female | 183 (30.5) |
| Religion | Christianity | 535 (89.2) |
| | Islam | 19 (3.2) |
| | Traditional | 32 (5.3) |
| | Others | 14 (2.3) |
| Education | Non-formal | 54 (9.0) |
| | Primary | 49 (8.2) |
| | Secondary | 161 (26.8) |
| | Tertiary | 336 (56.0) |
| Marital status | Single | 371 (61.8) |
| | Married | 201 (33.5) |
| | Divorced | 13 (2.2) |
| | Widowed | 15 (2.5) |
| Employment status | Employed | 311 (51.8) |
| | Unemployed | 284 (47.3) |
| | Retired | 5 (0.8) |

both districts (Fig 2A), but the difference was not significant ($\chi^2$ = 0.226, p = 0.634). The difference in fungi overall prevalence between Enugu (86.0%; 43/50) and Nsukka (94.0%; 47/50) was not significant ($\chi^2$ = 0.094, df = 1, p = 0.760) (**Table 3**).

The hands of five individuals in Nsukka were co-contaminated by at least two fungi species. *Aspergillus* spp. was the most commonly co-contaminating fungi on the palms of the commercial transport users; and was present in 80% of all cases of co-contaminations (**Table 4**).

Bacteria contaminated the hands of 40 (80.0%) and 47 (94.0%) of commercial transport users in Nsukka and Enugu respectively, giving a total contamination prevalence of 87%. The bacteria isolates were altogether 21 species, though it was difficult to get pure isolates of some bacteria. The four most prevalent bacteria species were *Escherichia coli* (16%), *Enterobacter* spp. (10%), *Proteus* spp. (8.0%), and *Stapylococcus aureus* (8.0%). The within district difference in bacteria species prevalence for Enugu was significant ($\chi^2$ = 20.519, df = 10, p = 0.025) (**Table 5**). Among the 21 bacteria species isolates, 11 (52.4%) were from Enugu and 15 (71.4%) were from Nsukka, while 5 (23.8%) were common to both districts (Fig 2B). The disparity in number of contaminating bacterial species between both districts was not significant ($\chi^2$ = 0.381, p = 0.537). The difference in prevalence of bacterial contamination between Enugu and Nsukka was not significant (47[94.0%] vs. 40[80.0%], $\chi^2$ = 0.302, df = 1, p = 0.583). No developmental stages of any parasite were isolated from the hand wash samples of the 22 commercial transport users who agreed to the handwashing.

Among the 21 fungi species isolates, 15 (71.4%) occurred in three or fewer numbers of the ten motor parks (**Fig 3**). None of the fungi isolates occurred in all ten parks. *Aspergillus* spp. and *Rhizopus* spp. occurred in all five parks in Nsukka (Park 6 –Park 10). Only *Malassazia* spp. occurred in as many as seven motor parks, while *Mucor* spp. occurred in six.

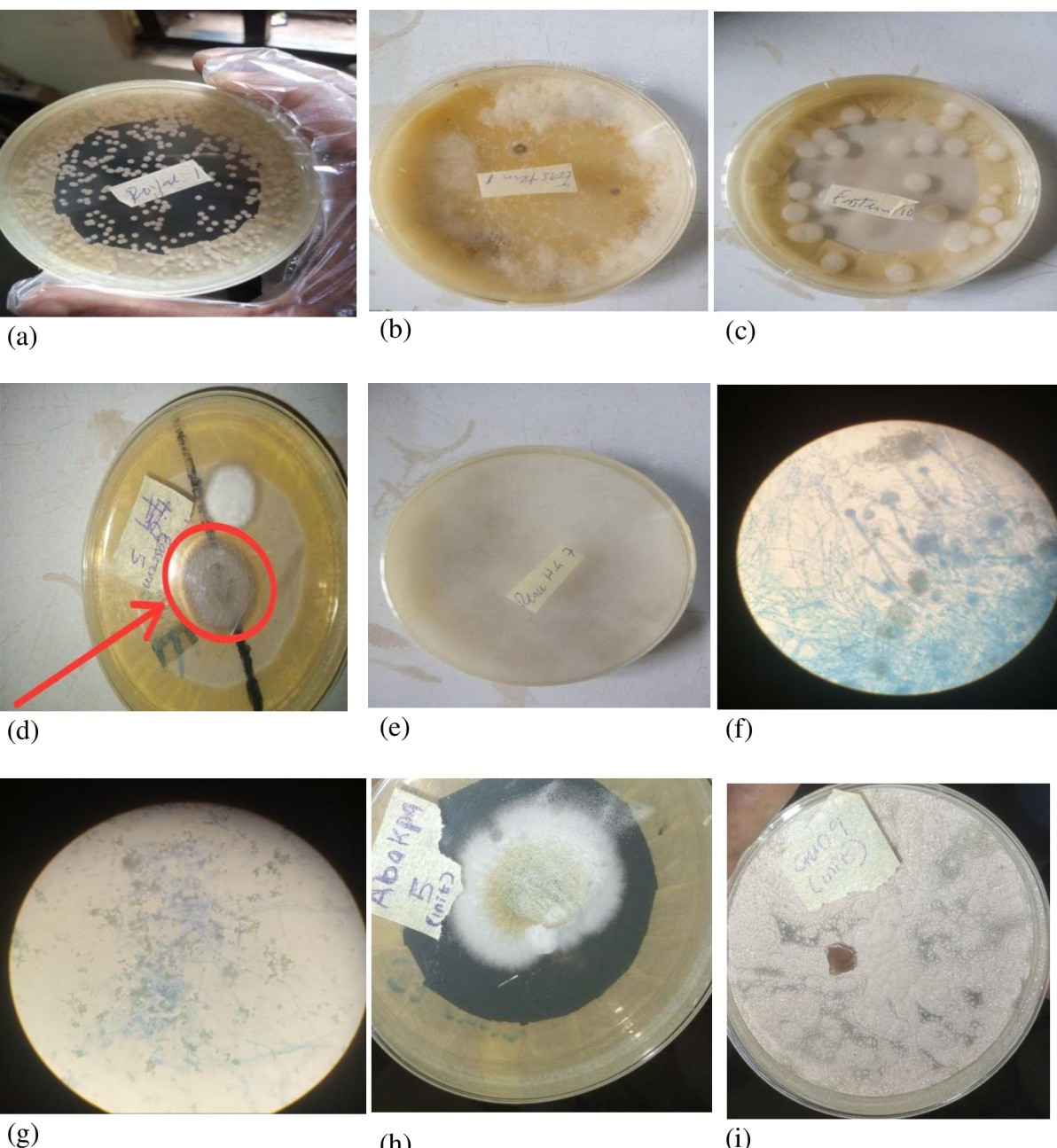

**Fig 1. Fungi isolates from commercial transport users in Enugu State, Nigeria.** Key: (a) *Malassazia* spp.; (b) *Thermothelomyces thermophiles*; (c) *Candida albicans*; (d) *Trichophyton rubrum*; (e) *Mucor* spp.; (f) *Trichosporon asahii* (x40); (g) *Trichophyton rubrum* (x40); (h) *Aspergilus flavus*; and (i) *Phaeoacrenium parasiticum*.

The bacteria, *E. coli* was isolated in all ten but one of the parks, Park 5. *Proteus* spp. was isolated in Park 1 –Park 5 which were all in Enugu town, and none from Nsukka town. More than 80% of the bacteria species isolates occurred in only three or fewer number parks (**Fig 4**).

## Questionnaire responses

**Hand hygiene knowledge.** All but four of the respondents, 596 (99.3%) considered hand washing as part of personal hygiene; 555 (92.5%) believed parasites/germs can be found on

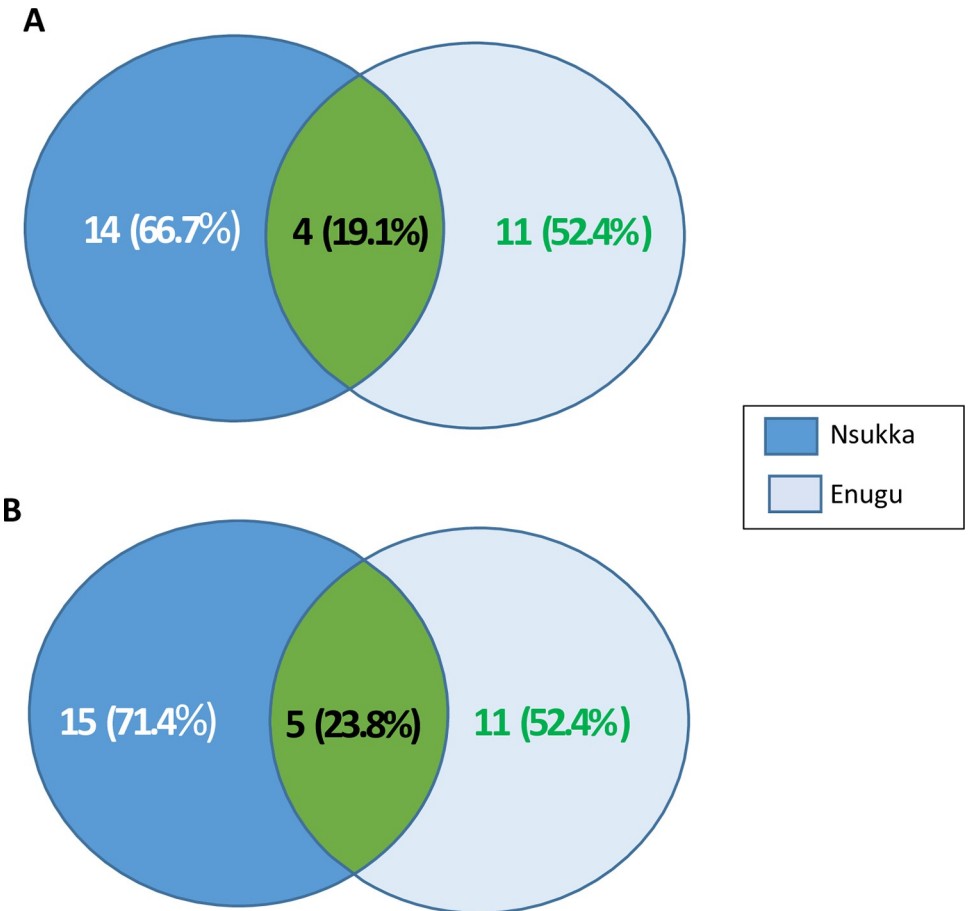

**Fig 2. Venn diagram ondifference in microbial diversity on hands of commercial transports in Nsukka and Enugu towns,Enugu State, Nigeria.** (A) Fungi; (B) Bacteria.

contaminated hands, and 489 (81.5%) believed coronavirus can be transmitted by poor hand hygiene. Those who believed that drivers and passengers encounter a lot of contaminated surfaces which make them as vulnerable to infections as health care workers were 522 (87.0%) (**Table 6**). The sources of germs responsible for hand contamination in the parks according to the responses given are presented in **Fig 5**.

**Beliefs about hand hygiene.** The hand hygiene beliefs of the commercial transport users are summarised in **Table 7**. In response to the query, "I am protecting my health by washing my hands or using hand sanitizer", 239 (39.8%) and 236 (39.3%), agreed and strongly agreed respectively. In response to the query, "I do not believe that shaking hands with strangers can get me contaminated with harmful microorganisms", 142 (23.7%) agreed, 114 (19.0%) were neutral, while 123 (20.5%) strongly disagreed, and 184 (30.7%) disagreed. Those who agreed that their hand hygiene practices increased greatly after the COVID-19 scare were 136 (22.7%). Those who were concerned about contacting COVID-19 were 53.5% ($n = 321$) (**Fig 6**).

Those who "always" wash their hands before eating or handling food were 382 (63.7%), and those who said they washed their hands "very often" before eating or handling food were 115 (19.2%). Less than 2.0% "rarely or never" washed their hands before eating or handling food items. After touching frequently used surfaces, 122 (20.3%) and 98 (16.3%) respectively, said

**Table 3. Fungi prevalence in hands of commercial transport vehicle users in Nsukka and Enugu town, Enugu State, Nigeria.**

| Fungi | Prevalence (%) | | | Pathogenicity |
|---|---|---|---|---|
| | Overall (n = 100) | Enugu (n = 50) | Nsukka (n = 50) | |
| *Aspergillus flavus* | 2 (2.0) | 2 (4.0) | - | OP [41,42] |
| *Aspergillus niger* | 1 (1.0) | - | 1 (2.0) | OP [43,44] |
| *Aspergillus* spp. | 14 (14.0) | - | 14 (28.0) | OP, P [44] |
| *Bipolaris* spp. | 6 (6.0) | - | 6 (12.0) | P [45] |
| *Blastomyces* spp. | 2 (2.0) | - | 2 (4.0) | P [46] |
| *Candida albicans* | 11 (11.0) | 9 (18.0) | 2 (4.0) | OP [47] |
| *Candida* spp. | 3 (3.0) | - | 3 (6.0) | OP [48] |
| *Malassezia* spp. | 14 (14.0) | 12 (24.0) | 2 (4.0) | NP [49] |
| *Microsporum gullinae* | 3 (3.0) | 3 (6.0) | - | NP [50] |
| *Mucor* spp. | 13 (13.0) | 7 (14.0) | 6 (12.0) | OP [51] |
| *Paecicomyces variotii* | 2 (2.0) | 1 (2.0) | 1 (2.0) | OP [52] |
| *Penicillum* spp. | 4 (4.0) | - | 4 (8.0) | NP, OP [53,54] |
| *Phaeoacremonium parasiticum* | 1 (1.0) | 1 (2.0) | - | P*, OP [55,56] |
| *Rhizopus* spp. | 6 (6.0) | - | 6 (12.0) | OP [57] |
| *Saccharomyces* spp. | 4 (4.0) | - | 4 (8.0) | OP [58] |
| *Thermothelomyces thermophiles* | 2 (2.0) | 2 (4.0) | - | OP [59] |
| *Trichoderma viride* | 1 (1.0) | - | 1 (2.0) | P* [60] |
| *Trichophyton* spp. | 1 (1.0) | - | 1 (2.0) | OP [61] |
| *Trichosporon asahii* | 1 (1.0) | 1 (2.0) | - | P, OP [62] |
| Unidentified species 1 | 3 (3.0) | 3 (6.0) | - | |
| Unidentified species 2 | 2 (2.0) | 2 (4.0) | - | |
| No growth | 10 (10.0) | 7 (14.0) | 3 (6.0) | |
| | 90 (90.0) | 43 (86.0) | 47 (94.0) | |
| | | $\chi^2$ = 38.549, df = 10, p < 0.0001 | χ2 = 46.069, df = 13, p < 0.0001 | |

OP–opportunistic pathogen; P–pathogenic; NP–non-pathogenic or mostly non-pathogenic; P*—pathogenic in plants.

they "always" and "very often" washed their hands. While 193 (32.2%), 151 (25.2%), and 36 (6.0%) said respectively that they "sometimes", "rarely", and "never" washed their hands after touching frequently used surfaces. After shaking hands with someone, 188 (31.3%) said they "rarely", while 175 (29.2%) said they never washed their hands. After removing their mask, 175 (29.2%) "rarely" washed their hands, while 213 (35.5%) never washed their hands (**Table 8**).

In response to the question, "How long does my hand washing take?" 276 (46.0%) of the respondents chose 10 s, 175 (29.2%) chose 15 s, 92 (15.3%) 5 s, and 57 (9.5%) 20 s (**Fig 7A**). After hand washing, 423 (70.5%) dry their hands with a towel/cloth, 95 (15.8%) dry on their clothes, and 62 (10.8%) do not dry (**Fig 7B**).

**Table 4. Fungi co-contamination in hands of commercial transport vehicle users in Nsukka, (*n* = 50).**

| Fungi co-contamination | Frequency (%) |
|---|---|
| *Aspergillus* spp., *Penicillum* spp. | 1 (2.0) |
| *Aspergillus* spp., *Bipolaris* spp. | 2 (4.0) |
| *Rhizopus* spp., *Bipolaris* spp. | 1 (2.0) |
| *Aspergillus* spp., *Bipolaris* spp., *Penicillum* spp. | 1 (2.0) |

**Table 5. Bacteria prevalence in hands of commercial transport vehicle users in Nsukka and Enugu town, Enugu State, Nigeria.**

| Bacteria | Prevalence (%) | | | Pathogenicity |
|---|---|---|---|---|
| | Overall (*n* = 100) | Enugu (*n* = 50) | Nsukka (*n* = 50) | |
| *Actinobacter* spp. | 2 (2.0) | - | 2 (4.0) | OP [63] |
| *Bacillus* spp. | 2 (2.0) | - | 2 (4.0) | OP [64] |
| *Citrobacter* spp. | 6 (6.0) | 6 (12.0) | - | OP [65] |
| *Clostridium difficile* | 2 (2.0) | - | 2 (4.0) | OP [66] |
| *Enterobacter* spp. | 10 (10.0) | 4 (8.0) | 6 (12.0) | OPr [67] |
| *Escherichia coli* | 16 (16.0) | 10 (20.0) | 6 (12.0) | NP, OP, P [68] |
| *Klebsiella pneumonia* | 1 (1.0) | - | 1 (2.0) | OP [69,70] |
| *Klebsiella* spp. | 5 (5.0) | 5 (10.0) | - | OP [71] |
| *Proteus mirabilis* | 1 (1.0) | - | 1 (2.0) | OP [72] |
| *Proteus* spp. | 8 (8.0) | 8 (16.0) | - | OP [73] |
| *Proteus vulgaris* | 2 (2.0) | - | 2 (4.0) | OP [74] |
| *Providencia* spp. | 3 (3.0) | 3 (6.0) | - | OP [75] |
| *Pseudomonas aeruginosa* | 1 (1.0) | - | 1 (2.0) | OP [76] |
| *Salmonella enterica* | 1 (1.0) | 1 (2.0) | - | P [77] |
| *Salmonella* spp. | 6 (6.0) | 3 (6.0) | 3 (6.0) | P [78] |
| *Serratia marcescens* | 3 (3.0) | 1 (2.0) | 2 (4.0) | OP [79] |
| *Shigella* spp. | 3 (3.0) | - | 3 (6.0) | P [80] |
| *Stapylococcus aureus* | 8 (8.0) | 4 (8.0) | 4 (8.0) | OP [81] |
| *Streptococcus* spp. | 3 (3.0) | - | 3 (6.0) | OP [82,83] |
| *Vibrio cholera* | 2 (2.0) | - | 2 (4.0) | OP [84] |
| *Yersinia* spp. | 2 (2.0) | 2 (4.0) | - | NP, OP [85] |
| No growth | 13 (13.0) | 3 (6.0) | 10 (20.0) | |
| | 87 (87.0) | 47 (94.0) | 40 (80.0) | |
| | | $\chi^2$ = 20.519, df = 10, p = 0.025 | $\chi^2$ = 14.196, df = 14, p = 0.435 | |

OP–opportunistic pathogen; OPr–rarely opportunistic pathogen; P–pathogenic; NP–non-pathogenic or mostly non-pathogenic; P*—pathogenic in plants.

Some of the most common reasons why they failed to wash their hands were due to forgetfulness 164 (27.3%), unavailability of water 143 (23.8%), lack of time 129 (21.5%), and laziness 114 (19.0%) (**Table 9**). Only 271 (45.2%) of the commercial transport users said they sometimes used hand sanitizers when outside the home.

The odds of good purposive hand hygiene practice appeared to decrease with age among the commercial transport users. The odds of good purposive practice was 44% less (OR = 0.557 (95% CI = 0.340–0.912), p = 0.020), and 73% less (OR = 0.266 (95% CI = 0.149–0.475), p < 0.001) among park users 30–39 years and 40 years and above respectively, compared to those 10–19 years. The odds of good practice increased with the level of education. Those who had tertiary (OR = 5.087, 95% CI = 2.535–10.208, p < 0.001) or secondary (4.372, 95% CI = 2.105–9.081, p < 0.01) education had five and four times higher odds respectively, of good purposive hand hygiene practice compared to those who had no formal education. Religion, employment status and marital status are other demographic variables that had significant relationship with purposive hand hygiene practice from crude odd ratio estimates (Table 10).

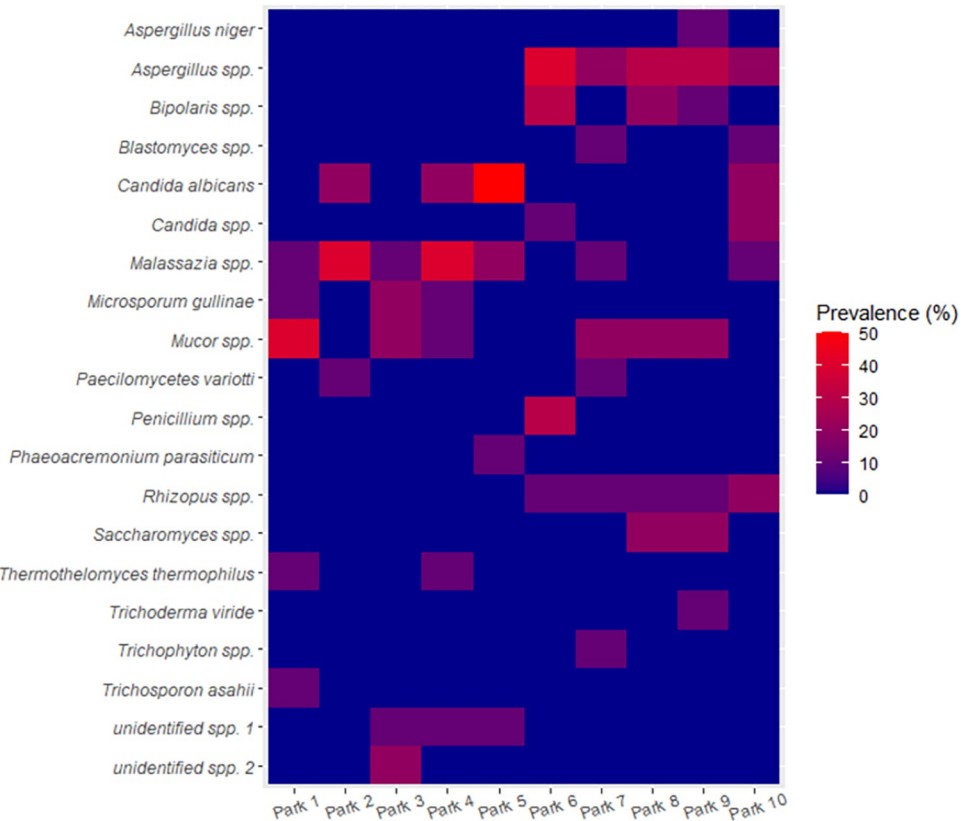

**Fig 3. Prevalence of fungi contamination on hands of commercial transport users in ten motor parks in Enugu State, Nigeria.**

## Discussion

This study showed a high prevalence of fungal (90.0%) and bacterial (87.0%) contamination among the commercial transport users in Enugu State, Nigeria. Microbial diversity was higher in Nsukka than Enugu probably due to the more diverse nature of the parks location in Nsukka. All the parks in Enugu town were located in urban areas, while two of the parks in Nsukka were in suburban areas. The fungal isolates from this study included pathogenic, opportunistic and non-pathogenic strains. The pathogenic fungi include *AspergiIllus* spp., which is responsible for the fungal infection known as aspergillosis; *Rhizopus* spp., which can cause mucormycosis; *Bipolaris* spp. may occasionally lead to phaeohyphomycosis, while some of the non-pathogenic fungi found in this study include, *Malassezia* spp., *Saccharomyces* spp., and *Mucor* spp. *Aspergillus* spp. was most prevalent fungus in this study, at 28.0% prevalence in the commercial transport users in Nsukka. This finding aligns with a study conducted by Osei *et al*. [86], at the University of Cape Coast Metropolis (UCC), Ghana, which reported *Aspergilus* spp. as the dominant fungi in the hand palm swab samples of 21 commercial drivers. In another study conducted by Keri *et al*. [87], to determine fungal carriage on healthcare workers' hands in a tertiary care centre in India, hand washes were collected from 60 health care workers and infused in brain heart infusion broth with gentamycin. The result showed that *Candida* spp. was the most prevalent fungi found on the palms of those health care workers. This is in contrast to the present study. The method of sample collection and culturing, as well as the sample population are different from the present study; while the present study

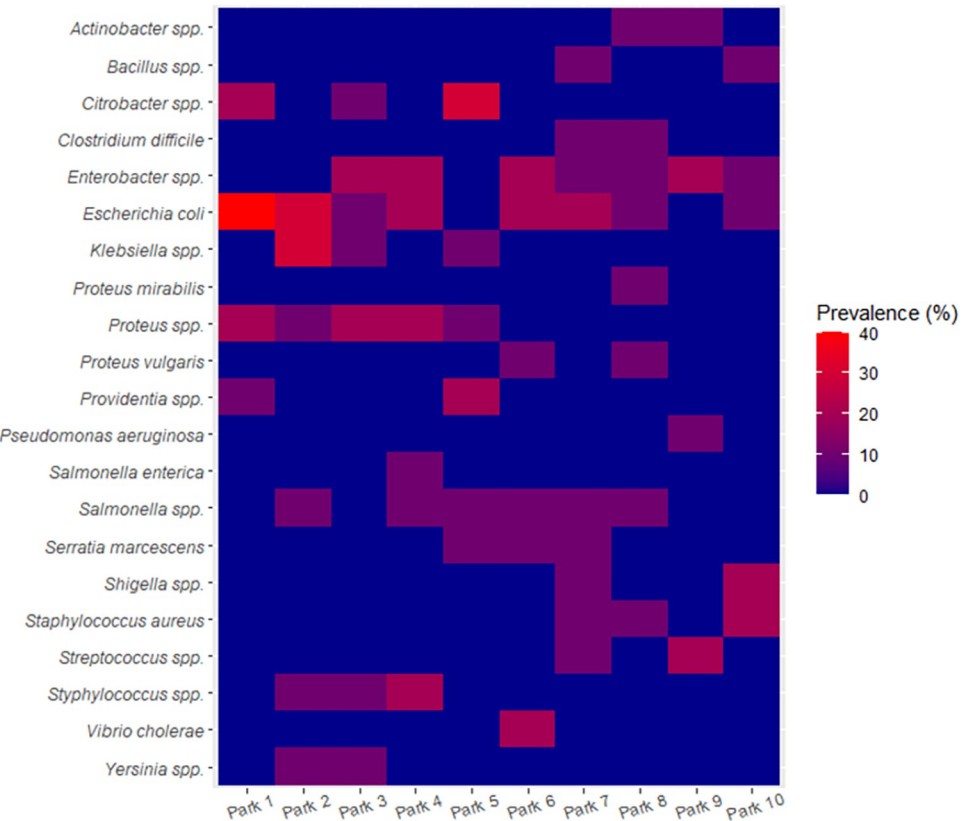

**Fig 4. Prevalence of bacteria contamination on hands of commercial transport users in ten motor parks in Enugu State, Nigeria.**

sampled commercial transport users, theirs was on health care workers. According to the study done by Dubljanin *et al.* [88] in Serbia, from 50 transport users; the prevalence of *Aspergillus* spp. was high among the transport users. Their finding on dominant prevalence of *Aspergillus* spp. is corroborated by the present study. In Enugu town, *Aspergillus* spp. was less prevalent fungi, *Aspergillus flavus* was only isolated from the palms of two individuals, 2 (4.0%). *Malassazia* spp. was the most prevalent fungi species (24.0%), a fungus naturally found on the skin surfaces of humans and many animals and is naturally harmless [89]. In some cases, they can be opportunistic pathogens causing skin conditions such as seborrheic dermatitis, dandruff and pityriasis versicolor [89]. *Candida albicans*, an opportunistic pathogenic yeast that causes the human infection, candidiasis [90], also had a high prevalence in Enugu town.

**Table 6. Knowledge of hand hygiene among commercial transport users of motor parks in Nsukka and Enugu towns, Enugu State, Nigeria (*n* = 600).**

| Knowledge | Frequency (%) |
| --- | --- |
| Hand washing is part of personal hygiene | 596 (99.3) |
| Parasites/germs can be found in contaminated hands | 555 (92.5) |
| Coronavirus can be transmitted by poor hand hygiene | 489 (81.5) |
| An alcohol-based hand sanitizer that contains 60% alcohol is sufficient for hands disinfectant | 392 (65.3) |
| Like healthcare workers, drivers and passengers do encounter a lot of contaminated surfaces that make them vulnerable to infections | 522 (87.0) |

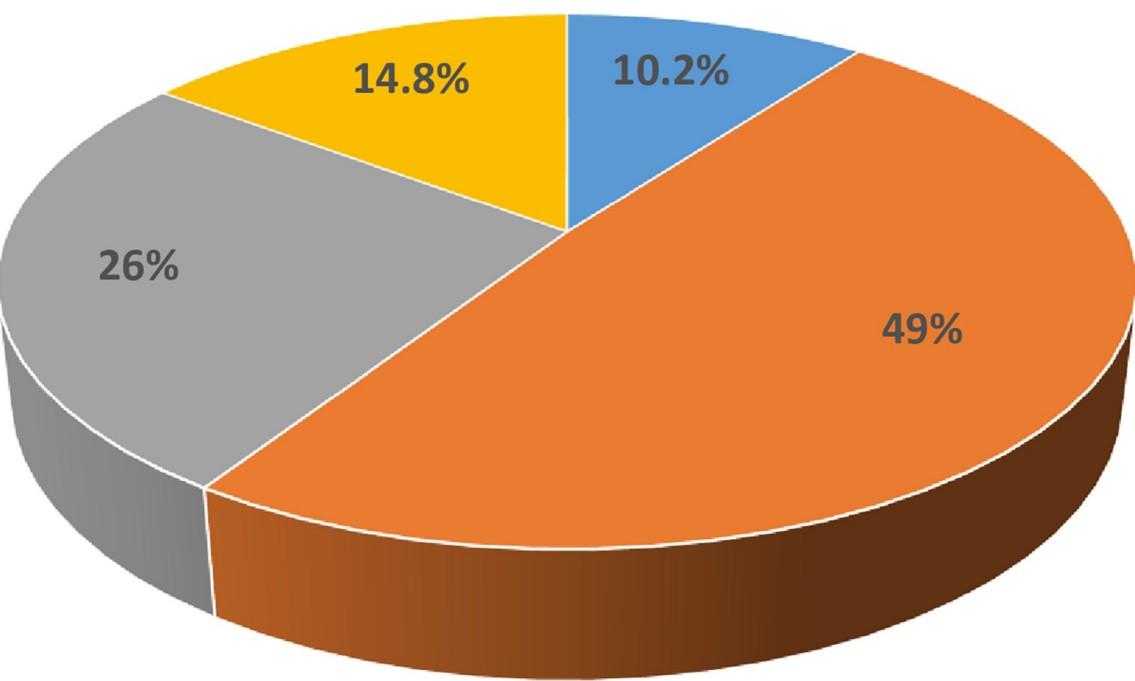

**Fig 5. Sources of germs responsible for hand contaminations in parks according to the response of commercial transport users of motor parks in Nsukka and Enugu town, Enugu State, Nigeria (*n* = 600).**

Fungal colonisation of the hands or other organs poses diverse risks. Fungal pathogens are linked to an estimated 13 million infections and 1.5 million deaths annually [91]. For examples, aside the direct infection and morbidities of *C. albican*, it has been linked to several

**Table 7. Hand washing beliefs among commercial transport users of motor parks in Nsukka and Enugu towns, Enugu State, Nigeria (*n* = 600).**

| Beliefs | Strongly disagree | Disagree | Neutral | Agree | Strongly agree |
|---|---|---|---|---|---|
| I am protecting my health by washing my hands or using hand sanitizer | 87 (14.5) | 19 (3.2) | 19 (3.2) | 239 (39.8) | 236 (39.3) |
| I do not believe that shaking hands with strangers can get me contaminated with harmful microorganisms | 123 (20.5) | 184 (30.7) | 114 (19.0) | 142 (23.7) | 37 (6.2) |
| I can get contaminated from harmful microorganisms if I touch my eyes, nose, or mouth with my unwashed hands, or not using hand sanitizer | 49 (8.2) | 61 (10.2) | 105 (17.5) | 218 (36.3) | 167 (27.8) |
| My hand hygiene practice increased greatly after COVID-19 | 35 (5.8) | 101 (16.8) | 155 (25.8) | 169 (28.2) | 140 (23.3) |
| Do you believe practicing hand washing can help you fight COVID-19 | 51 (8.5) | 54 (9.0) | 101 (16.8) | 208 (34.7) | 186 (31.0) |

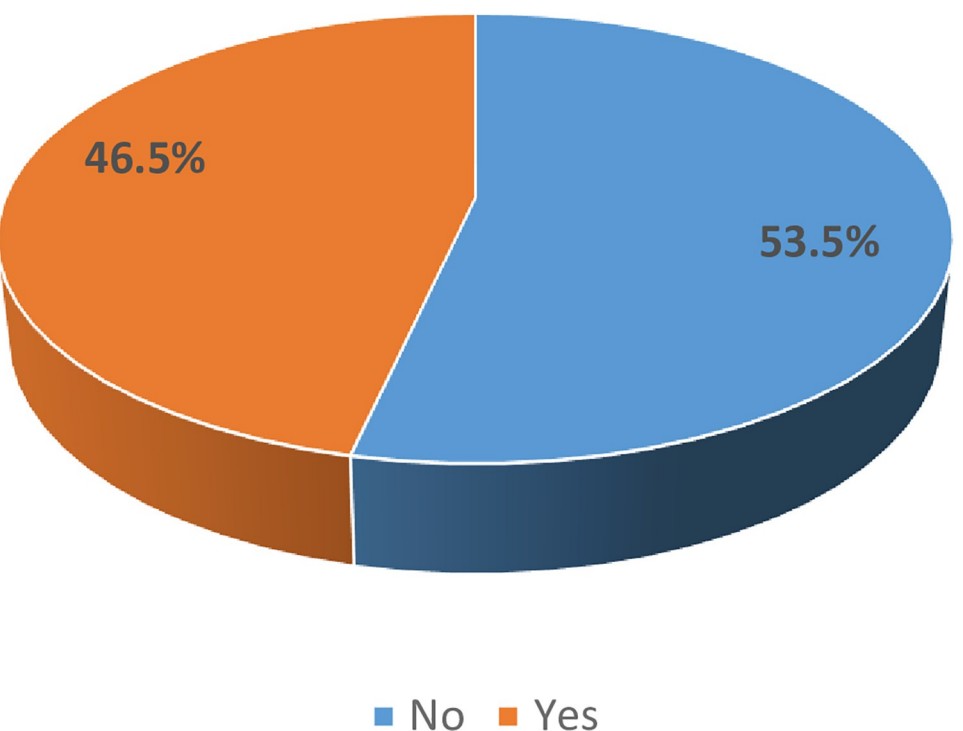

**Fig 6. Concerns about contacting COVID-19 among commercial transport users of motor parks in Nsukka and Enugu towns, Enugu State, Nigeria.**

pathological conditions such as diabetes, inflammatory bowel diseases (IBD), cancer, and metabolic diseases [92]. *Candida albican* and some bacteria (e.g. *Streptococcus* spp.) are known to synergistically interact to increase pathogenicity [93]. Fungi are able to serve as alternative host to viruses, including mammal, insect and plant associated fungi [94], suggesting a potential role in cross-transmission or spillover.

*Escherichia coli* and *Enterobacter* spp. were predominant bacteria species isolated from the study. *E. coli* the most prevalent bacteria in this study, is a gram-negative bacillus commonly found in the lower intestine of warm-blooded animals. While most strains are harmless, the pathogenic varieties cause gastroenteritis, meningitis, urinary tract infections, and septic shock in humans [95]. In the study by Osei et al. [86], *E. coli* was similarly the dominant species in

**Table 8. Hand hygiene practices among commercial transport users of motor parks in Nsukka and Enugu towns, Enugu State, Nigeria (*n* = 600).**

| Practices | Frequency (%) | | | | |
|---|---|---|---|---|---|
| I wash hands | Always | Very often | Sometimes | Rarely | Never |
| Before I eat or handle food | 382 (63.7) | 115 (19.2) | 92 (15.3) | 10 (1.7) | 1 (0.2) |
| After using a public toilet or urinal | 356 (59.3) | 173 (28.8) | 54 (9.0) | 17 (2.8) | 0 (0) |
| After blowing my nose, coughing, or sneezing | 216 (36.0) | 133 (22.2) | 180 (30.0) | 62 (10.3) | 9 (1.5) |
| After I touch frequently used surfaces | 122 (20.3) | 98 (16.3) | 193 (32.2) | 151 (25.2) | 36 (6.0) |
| When my hands are visibly dirty | 330 (55.0) | 130 (21.7) | 87 (14.5) | 31 (5.2) | 22 (3.7) |
| After shaking hands with someone | 77 (12.8) | 44 (7.3) | 116 (19.3) | 188 (31.3) | 175 (29.2) |
| After touching animals or pets | 151 (25.2) | 98 (16.3) | 123 (20.5) | 130 (21.7) | 98 (16.3) |
| After removing my mask | 69 (11.5) | 33 (5.5) | 110 (18.3) | 175 (29.2) | 213 (35.5) |

**A**

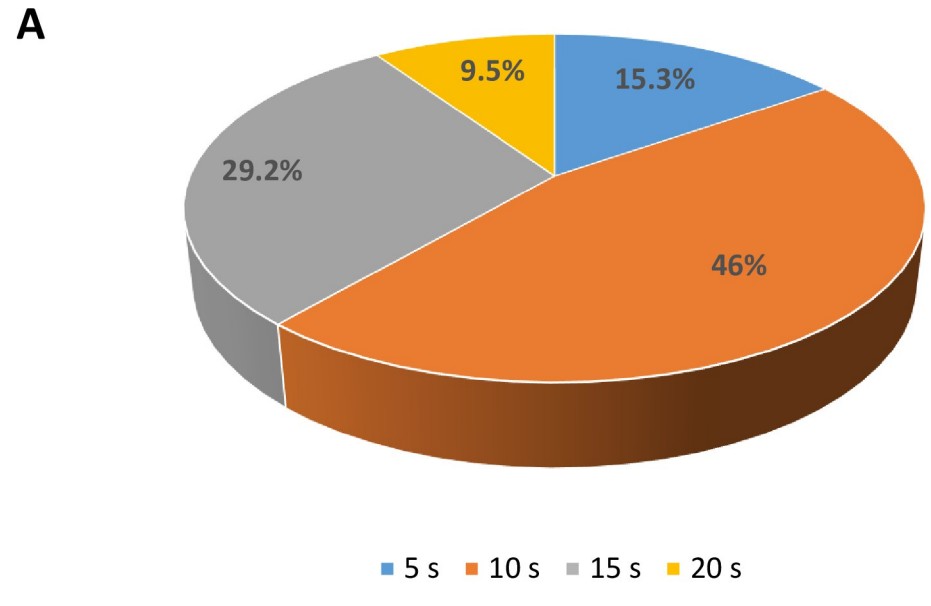

■ 5 s   ■ 10 s   ■ 15 s   ■ 20 s

**B**

After hand washing, I normally

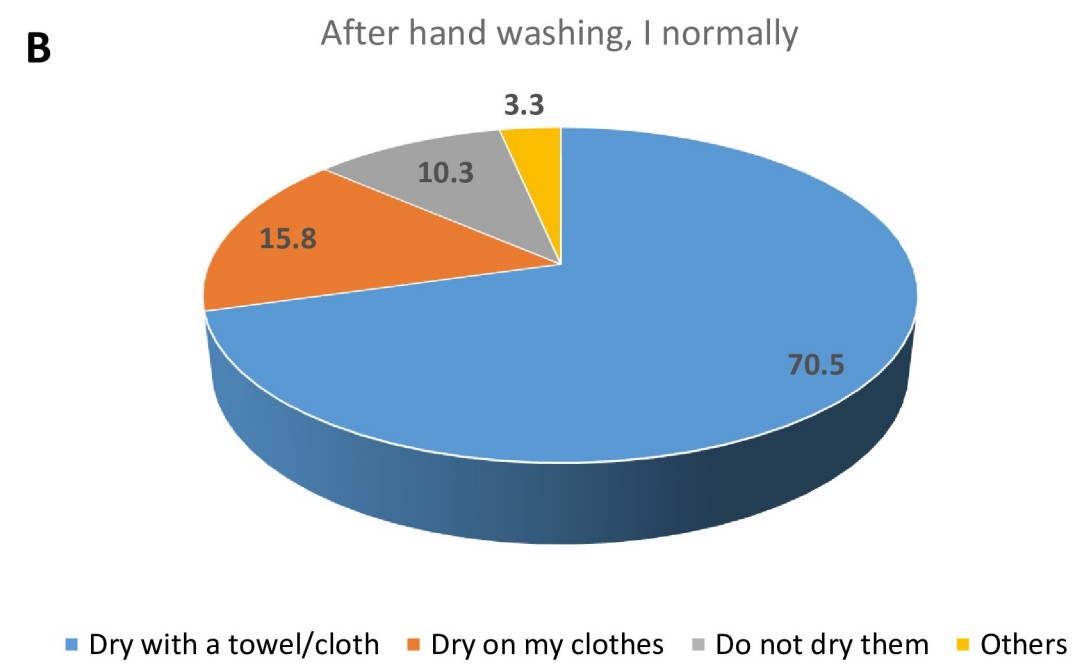

■ Dry with a towel/cloth   ■ Dry on my clothes   ■ Do not dry them   ■ Others

**Fig 7.** (A) Duration of hand washing, and (B) after hand washing routine among commercial transport users of motor parks in Nsukka and Enugu towns, Enugu State, Nigeria.

the palms of the drivers. The second predominant bacteria isolated from this study, *Enterobacter* spp., is pathogenic, and can cause respiratory tract infections, soft tissue infections and endocarditis. Some other bacteria species found in this present study includes *Staphylococcus aureus*, an opportunistic pathogenic bacteria responsible for over 1.1 million deaths in 2019 from a wide range of infectious syndromes including lower respiratory infections, bloodstream

**Table 9. Reasons for defaulting in hand washing among commercial transport users of motor parks in Nsukka and Enugu towns, Enugu State, Nigeria.**

|  | Options | Frequency (%) |
|---|---|---|
| The most common reasons for failing to wash ones hands | Laziness | 114 (19.0) |
|  | No available time | 129 (21.5) |
|  | Keep forgetting | 164 (27.3) |
|  | Soap is not available | 23 (3.8) |
|  | Water is not available | 143 (23.8) |
|  | Others | 27 (4.5) |

infections, meningitis, skin and subcutaneous infections, urinary tract infections, peritoneal and intra-abdominal infections, bone and joint infections, and cardiac infections [96]. Frequent occurrence of methicillin-resistant *S. aureus* (MRSA) which are resistant to antibiotics, and can be as high as over 50% in some countries, make the infections very problematic to manage and a leading cause of antimicrobial resistance attributed mortality [97–99]. Another is *Clostridium difficile* (also known as *Clostridioides difficile*), a pathogenic Gram-positive spore-forming and toxin-producing bacterium that can cause diarrhoea and inflammation of the colon. *C. difficile* is another major source of global public health concern, responsible for 15% - 25% of all cases of antibiotic-associated diarrhoea [100,101]. More recently, an increasing incidence of community acquired *C. difficile* infection indicates an evolving epidemiology from a majorly HAI to a community-acquired infection status [101]. *Salmomella* spp., is a pathogenic bacteria responsible for salmonellosis infection, which affects about 100 million people causing an estimated 59 000 to 155 000 deaths yearly, a situation worsened by

**Table 10. Crude odd ratio (OR) of purposive hand hygiene practice in relation to demographic variables of commercial transport users in Nsukka and Enugu towns, Enugu State, Nigeria.**

| Demographics | Options | OR (95% CI) | p |
|---|---|---|---|
| Age (years) | 10–19 | 1 |  |
|  | 20–29 | 1.171 (0.752–1.824) | 0.485 |
|  | 30–39 | 0.557 (0.340–0.912) | 0.020 |
|  | $\geq 40$ | 0.266 (0.149–0.475) | < 0.001 |
| Sex | Male | 1 |  |
|  | Female | 1.239 (0.875–1.755) | 0.228 |
| Religion | Christianity | 1 |  |
|  | Islam | 4.026 (1.319–12.289) | 0.014 |
|  | Traditional | 0.947 (0.464–1.936) | 0.882 |
|  | Others | 3.937 (1.086–14.270) | 0.037 |
| Education | Non-formal | 1 |  |
|  | Primary | 1.412 (0.564–3.531) | 0.461 |
|  | Secondary | 4.372 (2.105–9.081) | < 0.001 |
|  | Tertiary | 5.087(2.535–10.208) | < 0.001 |
| Marital status | Married | 1 |  |
|  | Single | 3.119 (2.172–4.479) | < 0.001 |
|  | Divorced | 7.135 (1.899–26.815) | 0.004 |
|  | Widowed | 1.070 (0.351–3.60) | 0.905 |
| Employment status | Employed | 1 |  |
|  | Unemployed | 2.255 (1.624–3.132) | < 0.001 |
|  | Retired | 0.367 (0.041–3.323) | 0.373 |

emergence of multi-drug-resistant (MDR) *Salmonella* serotypes [102,103]. *Proteus* spp., a Gram-negative opportunistic pathogenic bacteria. Three species of *Proteus*, *P. vulgaris*, *P. mirabilis* and *P. penneri* are responsible for many human urinary tract infections and kidney stones [104]. In a study by John and Adegoke [105] which evaluated hand contamination at bus terminals in Uyo metropolis, Nigeria, swab samples were taken using swab sticks and cultured in a nutrient agar media. Their result showed that *Bacillus* spp. was the dominant bacteria species which contrasts with the present study. In the present study, even though *Bacillus* spp. was isolated, it only occurred in 2.0% of the samples. Rayson et al. [106] identified *Enterobacter* spp. as the predominant bacterial contaminant on the hands of 20 sampled commercial park users in Mwanza region, Tanzania. The result agrees with findings from this study. The high contamination level of dominant hands in the present study shows a lack of proper hand hygiene practices among the commercial transport users, and a potentially high risk of spread of pathogenic life-threatening bacterial infections. Hand-to-face contacts and transfer to high-touch surfaces are potential routes of transmission of these pathogens both to self and others commercial transport users.

## Knowledge, belief, and practices

The findings in this study show that a very large number of commercial transport users have heard about hand hygiene and are aware that a good hand hygiene practice can help to curb various infections, but unfortunately, it remains one of the difficult activities for them to adopt as it is often difficult to remember to wash hands as often as required. In this study, a greater number of the respondents (99.3%) consider hand washing as a personal hygiene and were aware that germs are found on the hands, however this did not translate to high compliance rate. A study conducted by Natnael et al. [107] also showed that a greater number of taxi drivers (69.8%) in Dessie city, Ethiopia, had a good knowledge of hand hygiene but also had a low compliance rate. Wong and Lee [108] investigated the common missed areas during hand washing, 90.0% of the questionnaire respondents claimed to know that hand washing reduces contamination level of the hands by germ but only 40.0% adhered to the practice.

In this study, "After using the toilet" and "before eating" were the only instances where there was between 80% to 90% compliance to hand hygiene practices, which were habitual/customary hand washing practices. Other instances such as "after touching a pet" (41.5%), "after touching frequently used surface" (36.7%), "after handshake" (20.1%), "after blowing the nose" (58.2%), and after removing face mask (17.0%) which were purposive/deliberate hand washing practices had relatively low compliance rates. Curtis *et al.* [109] also reported that "after using urinary and defecating", "before eating or handling food materials", were the most common situations in which participants washed their hands. Pati et al. [110] reported that 86.7% of the respondents in their study washed their hands after using the toilet and before eating while the other situations recorded a low compliance rate. In a study done by Natneal et al. [111] to determine knowledge, attitude, and frequent hand hygiene practice during the COVID-19 pandemic in Makerere University, Katanga, only 24.5% of the questionnaire respondents had adequate knowledge of hand hygiene and a lesser number claimed to comply to hand hygiene practices. Additionally, from the crude odd ratio estimates, level of education, religion, employment status, and marital status were highlighted as factors that require further study as possible determinants of purposive hand hygiene practice among the commercial transport users.

For the duration of hand wash, a greater percentage of the respondents in this study claimed to complete their hand washing routine in 10 s. This contrasts with the study by Wong and Lee [108], where a greater percentage of the participants in their study completed

their hand washing routine in 20 s. Another study conducted by Borchgrevink et al. [111] showed that a greater percentage of the respondent washed their hands in less than 15s which can be said to agree with the present study. The reason for the variation in the duration of hand wash could be attributed to recall bias as people rarely time themselves when washing their hands, and guessing could introduce errors.

A greater percentage of participants in this study attributed their low practice of hand washing to forgetfulness (27.3%), unavailability of water (23.8%), and lack of time (21.5%) while fewer number attributed it to laziness (19.0%), lack of soap (3.8%) and other undisclosed reasons (4.5%). The study of Engdaw et al. [112] on the level of hand hygiene compliance in a health care centre in Ethiopia showed that unavailability of alcohol-based hand rub, adequate soap and water was the major reason for low compliance to hand hygiene practices. While this may suggest that provision of water and soap or hand sanitizers at strategic locations within commercial transport parks in Nsukka and Enugu towns could potentially encourage commercial transport users to practice better hand hygiene, a more robust strategic intervention would be required.

## Conclusion

From our study observation, we can conclude that the prevalence of bacteria and fungi hand contamination among commercial transport users in Enugu State is high. It can also be concluded that they have good knowledge of hand hygiene, but majority do not pay enough attention to practicing it. About 50% of the respondents seem to have reservations about the belief that poor hand hygiene practice can lead to the spread of COVID-19. A proper sensitization is needed in commercial parks on the mode of action of SAR-CoV-2, and other microbial and parasitic diseases, and the relationship between poor hand hygiene practice and their transmission. Also, public health policy makers should not assume that the emphasis on hand hygiene during the COVID-19 pandemic would automatically translate into improve hand hygiene compliance among commercial transport users post COVID-19 scare.

## Supporting information

**S1 Table. Varimax rotated principal component (PC) matrix of hand hygiene belief and practices among commercial transport users in Nsukka and Enugu towns, Enugu State, Nigeria.**
(DOCX)

**S2 Table. Classification matrix of purposive practice indicators (C4, C6, C7, C8) by the single generated binary purposive practice variable (Practice P50).**
(DOCX)

## Acknowledgments

The Department of Zoology and Environmental Biology and Department of Microbiology both in University of Nigeria, Nsukka. The Classic Biomedical Laboratory, Nsukka. The three provided some facilities used for conduct of the study.

## Author Contributions

**Conceptualization:** Ifeanyi O. Aguzie.

**Data curation:** Ifeanyi O. Aguzie, Ahaoma M. Obioha, Chisom E. Unachukwu.

**Formal analysis:** Ifeanyi O. Aguzie, Ahaoma M. Obioha, Chisom E. Unachukwu.

**Investigation:** Ifeanyi O. Aguzie, Ahaoma M. Obioha, Chisom E. Unachukwu, Kenneth O. Ugwu.

**Methodology:** Ifeanyi O. Aguzie, Ahaoma M. Obioha, Chisom E. Unachukwu, Onyekachi J. Okpasuo, Toochukwu J. Anunobi, Kenneth O. Ugwu, Patience O. Ubachukwu, Uju M. E. Dibua.

**Software:** Ifeanyi O. Aguzie.

**Supervision:** Ifeanyi O. Aguzie.

**Visualization:** Ifeanyi O. Aguzie.

**Writing – original draft:** Ifeanyi O. Aguzie, Ahaoma M. Obioha, Chisom E. Unachukwu, Onyekachi J. Okpasuo.

**Writing – review & editing:** Ifeanyi O. Aguzie, Ahaoma M. Obioha, Chisom E. Unachukwu, Onyekachi J. Okpasuo, Toochukwu J. Anunobi, Kenneth O. Ugwu, Patience O. Ubachukwu, Uju M. E. Dibua.

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
