## [Decision Letter · Decision Letter 0]

3 Jan 2024

PGPH-D-23-01961

Hand contamination and hand hygiene knowledge and practices among commercial transport users after the SARS-CoV-2 virus (COVID-19) scare, Enugu State, Nigeria

Dear Dr. Aguzie,

Thank you for submitting your manuscript to PLOS Global Public Health. After careful consideration, we feel that it has merit but does not fully meet PLOS Global Public Health’s publication criteria as it currently stands. Therefore, we invite you to submit a revised version of the manuscript that addresses the points raised during the review process.

We look forward to receiving your revised manuscript.

Kind regards,

Khadime Sylla

Academic Editor

Journal Requirements:

1. Please provide separate figure files in .tif or .eps format only and remove any figures embedded in your manuscript file. Please also ensure all files are under our size limit of 10MB.

Additional Editor Comments (if provided):

Please refer to reviewers comments in order to better present the manuscript.

Major revisions required.

Reviewers' comments:

Reviewer's Responses to Questions

**Comments to the Author**

1. Does this manuscript meet PLOS Global Public Health’s publication criteria? Is the manuscript technically sound, and do the data support the conclusions? The manuscript must describe methodologically and ethically rigorous research with conclusions that are appropriately drawn based on the data presented.

Reviewer #1: Yes

Reviewer #2: Yes

2. Has the statistical analysis been performed appropriately and rigorously?

Reviewer #1: I don't know

Reviewer #2: No

3. Have the authors made all data underlying the findings in their manuscript fully available (please refer to the Data Availability Statement at the start of the manuscript PDF file)?

Reviewer #1: Yes

Reviewer #2: Yes

4. Is the manuscript presented in an intelligible fashion and written in standard English?

Reviewer #1: Yes

Reviewer #2: Yes

5. Review Comments to the Author

Reviewer #1: this article meets the publication criteria of the journal. The english is done in a very good writing and is very clear. The methodology used on my knowledge alllows the results to be probant. However, for the sampling aspect of this study, an epidemiologist has to answer if the number of hand swab chosen is the one to do.

Reviewer #2: We consider this to be an interesting study that respected ethical rules. The recommendations from this study can have a positive impact at the community level. However, please find below some comments :

- Please, briefly describe the objectives in the abstract

- Please, briefly describe the statistical method for data analysis

- The results of the statistical comparison of proportions must be included in the abstract

- Comments should not be included in the results section (both in the summary and the full manuscript) . please just provide results

- In the introduction :

We think you can reduce the length of the text

The objectives must also be well worded (by including not only hand hygiene practices and hand contamination level among commercial park users, but also the comparison of contamination between the two cities)

- In the methodolgy section, please explain the method of randomization of the five parks and the participants from each park included in the study

- In the methodology section, line 55 : you should calculate the sample size using the SCHWARTZ formula and by maximizing the proportions p and p - 1 to 50% (the sample size will then be 384) and add the 10% of non-respondents. Wait until the results are written to enter the size sample of your study (600).

- int the methodology, line 155 : "non inclusion critaria" is more relevant than "exclusion critaria"

- in the results section : Confusion biais are possible in the comparison of microbial contamination between the two cities because we have no idea about how the participants included were randomized. It should be interesting to compare microbial contamination regarding demographic variables (sexe, level of education, marital statut, etc.) and Hand hygiene knowledge.

- avoid 3D figures

6. PLOS authors have the option to publish the peer review history of their article (what does this mean?). If published, this will include your full peer review and any attached files.

**Do you want your identity to be public for this peer review?** For information about this choice, including consent withdrawal, please see our Privacy Policy.

Reviewer #1: No

Reviewer #2: **Yes: **Moustapha DIOP

---

## [Decision Letter · Decision Letter 1]

9 May 2024

Hand contamination and hand hygiene knowledge and practices among commercial transport users after the SARS-CoV-2 virus (COVID-19) scare, Enugu State, Nigeria

PGPH-D-23-01961R1

Dear Mr Aguzie,

We are pleased to inform you that your manuscript 'Hand contamination and hand hygiene knowledge and practices among commercial transport users after the SARS-CoV-2 virus (COVID-19) scare, Enugu State, Nigeria' has been provisionally accepted for publication in PLOS Global Public Health.

Best regards,

Khadime Sylla

Academic Editor

Manuscript accepted in this version

Reviewer Comments (if any, and for reference):

Reviewer's Responses to Questions

**Comments to the Author**

1. If the authors have adequately addressed your comments raised in a previous round of review and you feel that this manuscript is now acceptable for publication, you may indicate that here to bypass the “Comments to the Author” section, enter your conflict of interest statement in the “Confidential to Editor” section, and submit your "Accept" recommendation.

Reviewer #1: All comments have been addressed

Reviewer #2: All comments have been addressed

2. Does this manuscript meet PLOS Global Public Health’s publication criteria? Is the manuscript technically sound, and do the data support the conclusions? The manuscript must describe methodologically and ethically rigorous research with conclusions that are appropriately drawn based on the data presented.

Reviewer #1: Yes

Reviewer #2: Yes

3. Has the statistical analysis been performed appropriately and rigorously?

Reviewer #1: Yes

Reviewer #2: Yes

4. Have the authors made all data underlying the findings in their manuscript fully available (please refer to the Data Availability Statement at the start of the manuscript PDF file)?

Reviewer #1: Yes

Reviewer #2: Yes

5. Is the manuscript presented in an intelligible fashion and written in standard English?

Reviewer #1: Yes

Reviewer #2: Yes

6. Review Comments to the Author

Reviewer #1: I had minors isuues and there have been mostly adressed by the authors

Reviewer #2: We are satisfied with the different responses from the authors. However, we are not asking to translate the tables into figures, but just the 3D form of the figures needs to be modified.

7. PLOS authors have the option to publish the peer review history of their article (what does this mean?). If published, this will include your full peer review and any attached files.

**Do you want your identity to be public for this peer review?** For information about this choice, including consent withdrawal, please see our Privacy Policy.

Reviewer #1: No

Reviewer #2: **Yes: **Moustapha DIOP
